# Comparison of hs-CRP in Adult Obesity and Central Obesity in Indonesia Based on Omega-3 Fatty Acids Intake: Indonesian Family Life Survey 5 (IFLS 5) Study

**DOI:** 10.3390/ijerph20186734

**Published:** 2023-09-08

**Authors:** Ginna Megawati, Noormarina Indraswari, Alexandra Aurelia Johansyah, Capella Kezia, Dewi Marhaeni Diah Herawati, Dida Achmad Gurnida, Ida Musfiroh

**Affiliations:** 1Doctoral Study Program, Faculty of Medicine, Universitas Padjadjaran, Bandung 40161, Indonesia; ginna@unpad.ac.id; 2Department of Public Health, Faculty of Medicine, Universitas Padjadjaran, Sumedang 45363, Indonesia; 3Medical Undergraduate Program, Faculty of Medicine, Universitas Padjadjaran, Sumedang 45363, Indonesia; 4Department of Child Health, Faculty of Medicine, Hasan Sadikin Hospital, Universitas Padjadjaran, Bandung 40161, Indonesia; 5Department of Pharmaceutical Analysis dan Medicinal Chemistry, Faculty of Pharmacy, Universitas Padjadjaran, Sumedang 45363, Indonesia

**Keywords:** hs-CRP, obesity, central obesity, omega-3 fatty acid intake, inflammations, IFLS 5

## Abstract

Obesity and central obesity are associated with dire conditions, such as metabolic syndrome, in which low-grade inflammation plays a part. C-reactive protein (CRP) is an inflammatory marker found to be elevated in those conditions. Omega-3 fatty acids work against inflammation and lower CRP levels in obese individuals. This study compared high-sensitivity CRP (hs-CRP) in adult obesity and central obesity in Indonesia based on omega-3 fatty acid intake using Indonesian Family Life Survey (IFLS) 5 data. Secondary data from household questionnaires were obtained from the IFLS 5 online database. Data from 3152 subjects were used; 76.65% of the subjects were female, with a mean age of 45.27 ± 15.77 years. Subjects were classified into five modified categories of obesity and central obesity based on body mass index (BMI) and waist circumference (WC). Omega-3 fatty acid intake was categorized into “low” and “adequate” based on dietary recommendations from the Mediterranean Diet Foundation (2011). There is a significant difference in hs-CRP based on modified obesity categories (*p* < 0.05). There was no significant difference in hs-CRP between low and adequate omega-3 intake (*p* > 0.05). These data suggest that hs-CRP is related to overweight, obesity, and central obesity. Meanwhile, omega-3 fatty acids are unrelated to hs-CRP. Further studies are needed to confirm these results.

## 1. Introduction

Obesity has nearly tripled since 1975 according to the World Health Organization (WHO). The number of overweight adults reached more than 1.9 billion in 2016, of whom 650 million were obese [1]. This trend also applies to developing countries, including Indonesia [2]. According to Riset Kesehatan Dasar (Riskesdas), the prevalence of adult obesity above 18 years old in Indonesia is 21.8% in 2018, which had increased since 2013 [3]. The diagnostic criteria for obesity are body mass index (BMI) ≥25 according to the WHO Asia-Pacific standard [4]. The diagnostic criteria of central obesity agreed upon by Kementerian Kesehatan Republik Indonesia (Kemenkes RI) are defined as a waist circumference (WC) of ≥90 cm for men and ≥80 cm for women [5]. Both obesity and central obesity have mainly been discussed in diabetes mellitus type 2 (DMT2), cardiovascular diseases, cancers, and metabolic syndrome [6,7,8,9,10].

The imbalance of energy intake and expenditure is the basis of obesity [6,10], supported by multifactorial aspects such as genetics, environment, psychosocial conditions, drugs, and hormones [6,11,12]. Abdominal fat accumulation, which leads to central obesity, has been hypothesized to occur due to the inability of subcutaneous tissue to collect more fat. This could be a consequence of genetic predisposition or stress. Visceral adipose tissue is known to be more active in terms of metabolism and lipolysis, which makes central obesity closely associated with a plethora of chronic diseases [13,14]. Excessive adipose tissue leads to several mechanisms which predispose comorbid conditions or complications. In obesity, the elevation of pro-inflammatory cytokines, decrease in adiponectin, and increase in lipolysis happen. These lead to insulin resistance, escalation of blood triglyceride levels, and blood pressure elevation, which are detrimental to the development of chronic diseases [9,13,15,16].

Low-grade chronic inflammation is one of the hallmarks of obesity and central obesity [17,18]. Pro-inflammatory mediators such as IL-6, IFN-γ, and TNF-α increase in obese and centrally obese individuals [19,20]. IL-6 stimulates hepatocytes to release C-reactive protein (CRP) [21]. Chronic inflammation may elevate cortisol levels and oxidative stress, which promote the development of metabolic syndrome, a cluster of conditions inducing various health problems [9,19,21,22]. Lifestyle modification, notably increased omega-3 fatty acid consumption, could counteract this problem. Omega-3 fatty acids are inversely related to inflammation [17], as they can prevent the release of pro-inflammatory cytokines through binding with the ligand for the G-protein-coupled receptor (GPR120) [23,24,25,26], which in turn decreases CRP levels [11,15,16,27]. Omega-3 fatty acids also reduce oxidative stress [15,28,29] and inhibit endothelial dysfunction [15,30,31,32,33].

The anti-inflammatory effect of omega-3 fatty acids, which could be a beneficial addition in managing obesity and central obesity, drives the initiation of this study. With data from the Indonesian Family Life Survey 5 (IFLS 5), this study sought to determine hs-CRP differences in various obesity groups and varied omega-3 fatty acid intake levels.

## 2. Materials and Methods

### 2.1. Subjects’ Characteristics

IFLS 5 is the fifth wave of an Indonesian socioeconomic and health longitudinal survey, conducted in 2014–2015. The respondents from 1993 represent 83% of Indonesia’s total population who resided in 13 different provinces [34]. Among numerous variables available, this study utilized sex, age, education, income, food consumption, weight, height, waist circumference (WC), and plasma equivalent levels of CRP.

Exclusion criteria included a history of chronic diseases such as hypertension, diabetes, tuberculosis, asthma, other lung conditions, heart attack, coronary heart disease, other heart problems, stroke, cancer or malignant tumor, arthritis or rheumatism, prostate illness, kidney disease, and stomach or other digestive disorders because chronic diseases increase CRP levels [35,36]. Thus, they are confounding factors in this study. Smoking and pregnancy status or history of childbirth less than one year from data collection were also excluded. CRP levels are also elevated in smoking [37,38] and pregnancy [39,40]. In addition, pregnancy would undoubtedly increase WC despite obesity or central obesity, and postpartum weight loss takes at least six months or more to return to weight before conception [41,42].

There were 7579 subjects whose CRP levels were observed. Subjects included were adults (≥19 years old) [43] whose sex, age, weight (kg), height (cm), WC (cm), and food consumption (fish, meat, eggs, and dairy) per week were recorded.

Subjects who had a smoking history (*n* = 2825), were pregnant or breastfeeding (*n* = 352), had a history of chronic diseases, or birth within less than one year of this study (*n* = 21) were excluded from this study. Subjects under 19 years old were not included (*n* = 541). Those who did not have sufficient or applicable data for BMI (*n* = 490) and those whose food frequency data for omega-3 fatty acid intake were insufficient or missing (*n* = 198) were also excluded from this study. Following inclusion and exclusion criteria, 3152 subjects remained suitable for this study (Figure 1).

The IFLS 5 survey and procedures have undergone ethical licensing assessments. They have been approved by the Institutional Board Review (IRB) in the United States and by the participating, Gadjah Mada University (UGM) in Indonesia. The ethical clearance number for IFLS 5 by RAND’s Human Subjects Protection Committee (RAND’s IRB) was s0064-06-01-CR01 [44]. Research Ethics Committee Universitas Padjadjaran Bandung approved the current study and released ethical clearance with the number 1339/UN6.KEP/EC/2022.

### 2.2. Laboratory Data of hs-CRP

Data were obtained from the IFLS 5 online database by measuring dried blood spot (DBS) values converted into CRP plasma equivalent values. However, the values generated may not represent those obtained from venous blood. Blood samples were obtained by finger prick on filter paper, which was then stored and processed in laboratory analysis using the enzyme-linked immunosorbent assay (ELISA) method [45].

### 2.3. Obesity and Central Obesity Assessments

Weight and height records were obtained from the book US with the code US06 for weight and US04 for height. BMI was calculated by weight (kg) divided by the square of height (m^2^) [6]. Waist circumference (WC) records were obtained in the same book with the code US06a [34,46]. The classification of obesity was based on the WHO Asia-Pacific standard [4], while the classification of central obesity followed the Kemenkes RI standard [5].

Subjects were divided into five categories based on BMI measurements: precisely normal, overweight, obese, centrally obese, and a combination of obese and centrally obese. Normal subjects had a BMI of <23, overweight subjects had a BMI of 23–24.9, and obese subjects had a BMI of ≥25, all with WC of <90 cm for men and <80 cm for women. Subjects with central obesity were those who exclusively had a WC of ≥90 cm for men and ≥80 cm for women. Subjects with both obesity and central obesity fell under the combination category with both a BMI of ≥25 and WC of ≥90 cm for men and ≥80 cm for women.

### 2.4. Omega-3 Fatty Acid Food Source Intake Measurements

The intake of omega-3 fatty acid food sources was based on four main food groups rich in omega-3 fatty acids: fish, meat, eggs, and dairy [47,48,49]. Food consumption frequency was recorded in book 3B with the code FM02B and FM03B for eggs, FM02C and FM03C for fish, FM02D and FM03D for meat, and FM02E and FM03E for dairy. The frequency noted was for the consumption of these food groups in one week before data collection started [46]. 

The Mediterranean diet standard performed categorization which supports high omega-3 fatty acid intake [50]. This study followed dietary recommendations from the Mediterranean Diet Foundation (2011) [51,52]. The recommended diet is an egg intake of at least two times per week, a fish intake of at least two times per week, a meat intake of at least two times per week, and a dairy intake of at least seven times per week or daily [51,52]. Subjects were considered to have “adequate” intake if they completely fulfilled the criteria. Otherwise, results were considered “low” intake.

### 2.5. Other Variables of the Study

Sex, age, education, and income were also included in this study. Sex was categorized into male and female with data obtained from form T2 with the code FT05 [46]. Age was the age from birth until the year of data collection, acquired from form T2 with the code FT06 [46] and categorized into three ranges, which are 19–44 (adults), 45–59 (pre-elderly), and ≥60 (elderly) [43]. Education was the last education level, obtained from book 3A with the code DL10, categorized into elementary, junior high, senior high, and D1, D2, and D3/university [46]. Income was noted in IDR based on monthly payment, obtained from book 3A with the code TK16a and TK16a1, categorized into <IDR 2 million, IDR 2–8 million, IDR 8–10 million, and >IDR 10 million [46]. 

### 2.6. Statistical Analysis

Pearson’s chi-squared and Fisher’s exact tests were used to determine independence or association between categorical variables. The Shapiro–Wilk test was used to determine the normality of the dataset, and Levene’s test was used to determine the homogeneity of variances. Normality assumptions were not met (*p* < 0.05), while homoscedasticity assumptions were met (*p* > 0.05). Therefore, nonparametric tests were used in this study. When comparing between more than two independent groups, the Kruskal–Wallis H test was performed, followed by the Mann–Whitney U test as post hoc testing if the previous test showed a significant difference. The Mann–Whitney U test was used when comparing two independent groups. The Bonferroni correction method was also performed. *p* < 0.05 was taken as the rejection value of the null hypothesis. Statistical analysis was conducted using Microsoft Excel 2019 (version 16.43, Microsoft Corporation, Redmond, WA, USA) and Stata/MP (version 14.1, StataCorp, College Station, TX, USA). 

## 3. Results

### 3.1. Study Population

Subjects’ demographics are presented in Table 1. There were more female subjects (76.65%) than male subjects (23.35%). The mean age was 45.27 ± 15.77 years. The age range of 19–44 had the highest number of subjects (*n* = 1591), making up 50.48% of the current study population. The prevalence of the age ranges had a decreasing trend with an increase in years.

Education data were only available for 442 subjects. Most subjects completed elementary education (34.84%), followed by a diploma or university education (33.26%). Income data were also only available for 161 subjects; most subjects had an income of less than IDR two million (84.71%). No subject was noted who had an income between IDR eight and ten million.

According to the modified obesity categories, 1005 (31.88%) were normal, 358 (11.36%) were overweight, 603 (19.13%) were obese, 397 (12.60%) were centrally obese, and 789 (25.03%) were both obese and centrally obese. The majority of subjects had normal BMI and WC. 

According to the Mediterranean diet standard, most subjects had low omega-3 fatty acid intake (96.29%). Only 117 subjects (3.71%) had adequate omega-3 fatty acid intake.

There were only 309 subjects whose fast food intake was accounted for. Subjects mostly had low physical activity (41.66%), followed by high physical activity (33.72%).

In Table 2, subjects’ demographics and data are further classified into the respective obesity categories. The hs-CRP values, shown as median (min.–max.), are also present here. Sex and obesity categories did not have an independent relation (*p* = 0.000). There was also a significant association between age and obesity categories (*p* = 0.000).

Among normal subjects, elementary education had the highest prevalence (43.59%). Senior high education was the highest in the overweight category (35.45%). For the rest of the categories, most of the subjects completed their education up to the diploma or university level (45.04% for the obesity category, 100% for the central obesity category, and 83.33% for the combination category). However, the education data for central obesity and combination categories were very limited, with only three and two data noted. 

Most subjects in the respective categories had an IDR two million or lower income. In the aforementioned income category, 44 (93.62%) were normal, 7 (41.18%) were overweight, 23 (92%) were obese, 22 (91.67%) were centrally obese, and 48 (84.21%) were both obese and centrally obese. Education and obesity categories had an independent relation (*p* = 0.617). 

Most subjects across all obesity categories had low omega-3 fatty acid intake. The overweight category had the fewest number of subjects with adequate omega-3 fatty acid intake (*n* = 8), while the combination category had the greatest number of subjects (*n* = 33). For subjects with low omega-3 fatty acid intake, the normal category had the highest number of subjects (*n* = 975), and the overweight category had the lowest (*n* = 350). There was a significant association between omega-3 fatty acid intake and obesity categories (*p* = 0.02).

As seen in Table 2, there was no significant association between fast food intake and obesity categories (*p* = 0.42). Physical activity and obesity categories also had no significant association (*p* = 0.96).

The median hs-CRP values had an increasing trend from normal, overweight, central obesity, obesity, and combination categories, in that particular order. Combination category had the highest median hs-CRP value at 0.17 (0.00–3.79) mg/dL. The normal category had the lowest median hs-CRP value at 0.05 (0.00–4.62) mg/dL.

In Table 3, subjects’ demographics and data are further classified by the respective omega-3 fatty acid intake levels. The hs-CRP values based on omega-3 fatty acid intake groups are also shown here as median (min.–max.). There was no association between sex and omega-3 fatty acid intake (*p* = 0.297). Age and omega-3 fatty acid intake also showed no association (*p* = 0.176).

Education and omega-3 fatty acid intake had an independent relation (*p* = 1.000), as did income and omega-3 fatty acid intake (*p* = 0.134). However, there was a significant association between obesity categories and omega-3 fatty acid intake (*p* = 0.020).

As shown in Table 3, fast food intake and omega-3 fatty acid intake had no significant relation (*p* = 0.46). There was also no significant association between physical activity and omega-3 fatty acid intake (*p* = 0.67).

The median hs-CRP values between those with adequate and low omega-3 fatty acid intake had a very small difference of 0.02 mg/dL. 

### 3.2. Comparison of hs-CRP Based on Obesity Categories

There was a significant difference in hs-CRP values between at least two different obesity categories (*p* = 0.0001). There was also an increasing trend from the normal BMI and WC (normal) category to higher BMI and WC (overweight, obesity, central obesity, and combination) categories.

The hs-CRP data were shown as median (min.–max.). The combination category had the highest hs-CRP value (median [IQR] = 0.17 (0.07–0.37) mg/dL), while the normal category had the lowest hs-CRP value (median [IQR] = 0.05 (0.02–0.14) mg/dL).

Table 4 shows statistical analysis by Kruskal–Wallis H test. Data are shown as median (min.–max.). The hs-CRP values based on obesity categories yielded a significant difference.

Following the significant results from the Kruskal–Wallis H test, a post hoc analysis using the Mann–Whitney U test was performed. Obesity and combination categories showed no significant difference in hs-CRP values (*p* = 0.43), while the other pairs of categories showed a significant difference (*p* < 0.05).

Table 5 shows post hoc analysis by Mann–Whitney U test. The hs-CRP values were all significantly different except between the obesity and combination categories.

### 3.3. Comparison of hs-CRP Based on Omega-3 Fatty Acid Intake Levels

Table 6 shows no significant difference when comparing hs-CRP values based on omega-3 fatty acid intake levels by the Mediterranean diet standard (*p* = 0.93). Those with adequate omega-3 fatty acid intake had a slightly higher hs-CRP value than those with a low omega-3 fatty acid intake (median [IQR] = 0.12 (0.04–0.25) mg/dL vs. 0.10 (0.04–0.26) mg/dL). 

## 4. Discussion

The current study aimed to compare the hs-CRP values between assorted obesity categories and different omega-3 fatty acid intake levels among 3152 subjects whose hs-CRP values were collected through the IFLS 5 survey. The particular secondary data used in this study measured CRP concentrations using the hs-CRP ELISA method. This method is more sensitive to low-grade inflammation and is, therefore suitable for this study. In this investigation, the results yielded a significant difference in hs-CRP values based on different obesity categories. However, there was no significant difference in hs-CRP values based on omega-3 fatty acid intake levels. Most subjects had low omega-3 fatty acid intake according to the Mediterranean diet standard. The discrepancy between the number of subjects who had adequate and low omega-3 fatty acid intake was also notably high (*n* = 117 vs. *n* = 3035). The previously mentioned result is in line with a study claiming that fish intake, which affects omega-3 levels in individuals, was very low compared to the vast amount of fish resources available among Indonesians [53].

Results showed a significant difference in hs-CRP values between subjects with normal BMI and WC and those with higher BMI and WC. Several previous studies also showed similar results. In a systematic review and meta-analysis by Choi et al. (2013), an increased odds of elevated CRP was related to obesity and overweight in all populations [54]. A study by Paepegaeya et al. (2015) showed that CRP was positively correlated with an increase in BMI and fat mass [55]. The pathophysiology linking obesity and elevated CRP levels has been widely discussed. Fat accumulation, especially in the abdomen, is tightly related to the development of enlarged and damaged adipose cells [9]. Adipose tissue is an endocrine organ that produces cytokines such as tumor necrosis factor (TNF), interleukin (IL)-6, IL-1, IL-18, and chemokines [6,21]. Moreover, free fatty acids activate serine kinase pro-inflammatory cascades, such as IkkapaB (IkB) kinase (IKK) and c-JunN-terminal kinase (JNK) [21,56]. These cascades induce the production of cytokines and stimulate the hepatic synthesis of CRP [21]. The transcription of CRP typically happens in hepatocytes as a response of cytokine elevation, particularly IL-6 [21,57]. Its effect is further amplified by IL-1 [57]. Therefore, obesity is considered a low-grade inflammatory disease [17,18].

Nonetheless, there were some limitations in this study that should be addressed. The data utilized were gathered manually as secondary data, which implies that there could have been human errors during measurement or data recording. Subject bias was also inevitable as these were data from questionnaires. The hs-CRP levels used were also plasma equivalent values converted from the dried blood spot (DBS) method results. Hence, the values may not entirely represent those taken directly from venous blood [45]. The hs-CRP values of subjects could also be influenced by factors in subjects’ lives that could not be considered in this study. Hence, the differences of hs-CRP values in this study may not be completely attributed to BMI and WC. Food choices or patterns could affect CRP concentrations. A systematic review by Defagó et al. (2014) aimed to find the correlation between food patterns and endothelial biomarkers. From these studies, it can be deduced that Westernized food patterns that are high in processed meats, sweets, fried foods, and refined grains showed a positive association with inflammatory markers such as CRP. Concurrently, healthy food patterns rich in fruits and vegetables had a more favorable influence on circulating CRP levels [58]. Food choices could also impact hs-CRP levels despite BMI and WC. 

Another possible confounder in this study is physical activity. The association between physical activity and CRP is also discussed in some studies. Loprinzi et al. (2013) enrolled US adults and children to examine the association between objectively measured physical activity with an accelerometer and CRP. It was found that physical activity was inversely correlated with CRP in adults but unrelated in children [59]. A cross-sectional study conducted by León-Latre et al. (2014) investigated the association between a sedentary lifestyle and inflammatory markers in healthy male workers. Workers with more sitting time had a worse inflammatory profile with higher CRP levels [60]. A review article by Michigan et al. (2011) analyzed the response of CRP levels to different exercise regimens in various age groups. The combination of diet and exercise programs appeared to have more significant effects than just physical activity alone, with substantial CRP reductions in five trials [61]. 

Psychosocial stress also affects CRP levels and is a confounding factor in this study. A systematic review by Johnson et al. (2013) evaluated the impact of chronic psychosocial stress on CRP levels. From 41 articles analyzing different types of stress, it was concluded that psychosocial stress significantly elevated CRP levels [62]. Negative emotions such as stress and depression may directly influence elevated pro-inflammatory cytokine production [63,64]. Another systematic review by Howren et al. (2009) assessed the correlation between depression and inflammatory markers such as CRP, IL-1, and IL-6. Clinically depressed patient samples showed the strongest positive correlations, though significance was also noted in community samples [64]. Stress-related behaviors such as sleep disturbances also showed a positive correlation to inflammation and, thus, higher CRP levels [65,66]. Cellular activation induced by stress could be mediated by the nuclear factor kappa-light-chain-enhancer of activated B cells (NF-κB). Through NF-κB activation, pro-inflammatory cytokines are released, increasing CRP [67,68].

Sex and age could affect CRP concentrations to a certain level. According to Lakoski et al. (2006), CRP levels were found to be higher in women than men despite confounders being accounted for [69]. A study by Wyczalkowska-Tomasik et al. (2015) stated that healthy older adults showed higher serum CRP levels compared to younger counterparts, though the values still fell below the normal range [70]. Wener et al. (2000) also found differences in CRP concentrations based on age and gender [71]. Genetic variation is also a factor that can influence low-grade inflammation, but there was no information on this from the utilized data. Galmés et al. (2019) stated that inflammation and omega-3 fatty acid response could be influenced by genetic variation [72]. A meta-analysis by Ligthart et al. (2016) discovered that many genetic loci were associated with chronic low-grade inflammation [73].

Although many studies claimed that omega-3 fatty acids have anti-inflammatory effects and are inversely related to inflammatory markers such as hs-CRP, this study showed no significant difference. The results of this study may be attributed to some confounding factors. Food frequency of large food groups were the only data available from IFLS 5. Food preparation, types of food, and the number of servings needed to be clearly defined. Specifically for fish, a study stated that there is more omega-3 fatty acid content in marine fish such as salmon, sea bass, cod, and barramundi than freshwater fish such as carp [74]. Different preparation methods could also alter the content of omega-3 PUFA in fish. The lowest amounts of omega-3 fatty acids could be seen in the deep-frying method preparation of New Zealand king salmon in a study conducted by Alsen et al. (2010) [75]. A survey by Nett et al. (2014) also showed that fried fish tend to have more omega-6 and MUFA content, possibly due to the canola oil used for frying [76]. Hence, the preparation, type, and number of servings of fish impact the amount of omega-3 fatty acids consumed.

A diet high in omega-6 PUFA could potentially be harmful instead, even contributing to the development of breast cancer, according to a few studies [77,78,79]. This could be due to competition between omega-3 and omega-6 fatty acids for cyclooxygenase (COX) and lipoxygenase (LOX) enzymes, which then affect the synthesis of eicosanoids. Eicosanoids participate in a few physiological processes, such as pro-/anti-inflammatory response, pro-/anti-platelet aggregation, vasodilation/vasoconstriction, immune response, and cell growth and development. Eicosanoids derived from omega-3 PUFA, such as eicosapentaenoic acids (EPA), have anti-inflammatory effects, while eicosanoids derived from omega-6 PUFA, such as arachidonic acids (AA) have pro-inflammatory effects. Consumption of omega-3 and omega-6 PUFA through food intake has a direct effect on tissue levels and eicosanoid metabolites of both PUFA [16,47,80]. Inflammation is linked to the release of cytokine production, which then affects the release of CRP from hepatocytes. Therefore, omega-6 could potentially increase CRP.

Besides fish, the types of dairy and meat consumed could also impact hs-CRP. Processed dairy products high in refined sugars and processed red meats would contribute to the unhealthy eating pattern associated with higher CRP levels, as mentioned previously [58]. It is also important to note that besides the four general groups of food consisting of eggs, fish, meat, and dairy, subjects consumed many other foods that might impact their food patterns and, thus, CRP levels. This anonymous information might also alter one’s adequacy status of omega-3 fatty acid intake since other foods such as seafood, nuts, seeds, certain oils, and grains are also high in omega-3 fatty acids [47,48,49].

Other than the factors mentioned above, the categorization of omega-3 fatty acid intake based on the Mediterranean diet standard also had its flaws. The complete dietary recommendation from Mediterranean Diet Foundation (2011) does not only consist of eggs, fish, meat, and dairy. It includes various other food groups such as olive oil, vegetables, fruits, bread and cereals, legumes, nuts, and red wine [51,52]. Consequently, the omega-3 fatty acid content analyzed in this study might not be wholly accurate, as consumption of other food might also increase omega-3 fatty acid intake and vice versa. However, only four food groups could be assessed due to data limitations.

Some previous studies also showed no association between increased omega-3 fatty acid intake and CRP levels. Madsen et al. (2003) conducted a prospective 60-person survey with omega-3 PUFA supplements. It was concluded that dietary supplementation of omega-3 fatty acids did not affect serum concentrations of CRP in healthy subjects [81]. Another prospective study conducted by Chan et al. (2002) investigated the effects of treatment with atorvastatin and fish oil on hs-CRP in individuals with visceral obesity. The decrease in plasma hs-CRP was seen with atorvastatin or atorvastatin with fish oil treatment but not with fish oil alone [82]. These two studies utilized fish oil supplements with more concentrated levels of omega-3 fatty acids than those obtained from a normal diet. There is also a prospective 21-person study by Mezzano et al. (2001) that compared Mediterranean and high-fat diets, supposedly high in omega-3 PUFA content, on plasma concentrations of hemostatic cardiovascular risk factors. There was no effect on CRP levels from either diet observed in healthy male individuals [83].

However, other studies contradicted this study’s results and showed positive correlations between omega-3 fatty acid intake and CRP levels. A survey by Saifullah et al. (2007) investigated the effects of 12-week fish oil supplementation on 27 hemodialysis patients in cardiovascular risk biomarkers. There was a reduction in both mean CRP levels and triglyceride levels in patients [84]. Another study by Mortazavi et al. (2018) enrolled 46 male patients with cardiovascular disease to observe the effects of omega-3 PUFA supplementation on serum apelin levels, hs-CRP, and lipid profiles. The supplementation of omega-3 fatty acids decreases serum hs-CRP levels. Tsitouras et al. (2008) also studied the influence of a high omega-3 fatty acid diet with 720 g of fatty fish weekly and 15 mL of sardine oil daily on healthy older adults. There was a notable reduction in serum CRP levels [85].

The anti-inflammatory effect of omega-3 fatty acids in relation to CRP could be through the inhibition of IKK and JNK phosphorylation, which in turn inhibits the NF-κB signaling process. EPA and docosahexaenoic acid (DHA) were found to be effective in inhibiting NF-κB through the activation of G-protein-coupled receptor 120 (GPR120) [15,24,25,26]. The IKK complex takes part in the signal transduction cascade of NF-κB, a protein complex that controls the production of cytokines and, thus, affects genes related to inflammation. Transcription of CRP takes part in hepatocytes as a response to elevated inflammatory cytokines such as IL-6 [57]. Therefore, inhibition of NF-κB by omega-3 fatty acids could decrease inflammatory cytokines and indirectly decrease CRP levels.

Another mechanism related to omega-3 fatty acids’ anti-inflammatory influence could be through the expression of peroxisome proliferator-activated receptor (PPAR)α/γ. PPARα activation inhibits NF-κB signaling, resulting in decreased production of inflammatory cytokines. In addition, the PPARγ ligand can inhibit macrophage activation and production of inflammatory cytokines, such as TNF-α, IL-6, and IL-1β [15,86,87]. Omega-3 fatty acids can also regulate leukotrienes, prostaglandins, and thromboxane synthesis [15]. Reducing inflammatory cytokines will reduce the release of CRP by hepatocytes, and CRP serum concentration will decrease.

Concurring with the discussion above, CRP and omega-3 fatty acids are important parameters for consideration in obesity because they are associated with inflammation and metabolic health, which play significant roles in the development and management of obesity. Several other factors could also influence obesity. However, the data utilized in this study, although they encompass a major portion of Indonesia, were unfortunately very limited in relevance to inflammation in obesity. Thus, the current study was only able to use hs-CRP and omega-3 data to investigate inflammation in obesity.

## 5. Conclusions

In conclusion, a significant difference in hs-CRP values was found between those who had normal BMI and WC compared to those who had higher BMI and WC. There was a notable increasing trend between those with normal BMI and WC and those with higher BMI and WC. Food choices, physical activity, and psychosocial stress are possible confounders in this study as they could affect hs-CRP concentrations. The result of no significant difference in hs-CRP values between adequate and low omega-3 fatty acid intake should be taken with a grain of salt, as the secondary data used in this study were limited in terms of food consumption. Food frequencies in general food groups were the only available data from IFLS 5 and, thus, categorization was also loosely based on the Mediterranean diet standard. 

Further research should focus on finding the influence of obesity and omega-3 fatty acids on hs-CRP concentrations while considering that many other factors that could alter hs-CRP. Utilization of food data containing detailed information on preparation methods, types of food, and the number of servings could be beneficial. This would allow a more accurate analysis, bringing more clinical relevance to practice and allowing optimal care for the population involved. 

## Figures and Tables

**Figure 1 ijerph-20-06734-f001:**
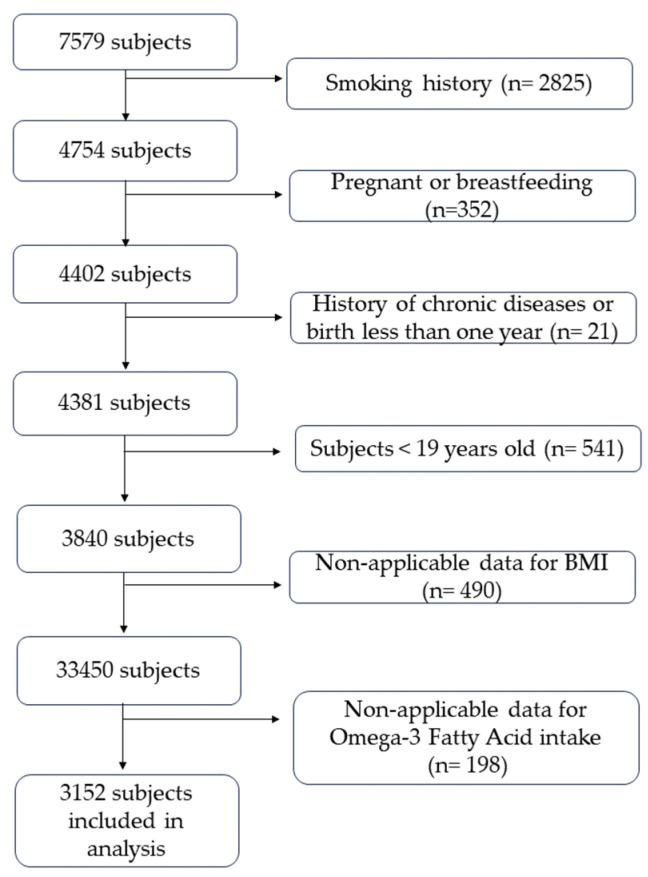
Inclusion and exclusion criteria.

**Table 1 ijerph-20-06734-t001:** Baseline characteristics of subjects.

Characteristics		Frequency (*n*)	Percentage (%)
Sex (*n* = 3152)	Male	736	23.35
	Female	2416	76.65
Age (*n* = 3152)	19–44	1591	50.48
[45.27 ± 15.77]	45–59	938	29.76
	≥60	623	19.77
Education (*n* = 442)	Elementary	154	34.84
	Junior high	28	6.33
	Senior high	113	25.57
	D1, D2, D3/university	147	33.26
Income (*n* = 161)	<IDR 2 million	144	84.71
	IDR 2–8 million	15	8.82
	IDR 8–10 million	0	0
	>IDR 10 million	2	1.18
Obesity categories	Normal	1005	31.88
(*n* = 3152)	Overweight	358	11.36
	Obesity	603	19.13
	Central obesity	397	12.60
	Combination	789	25.03
Omega-3 fatty acid intake	Adequate	117	3.71
(*n* = 3152)	Low	3035	96.29
Fast food intake	Seldom	182	58.90
(*n* = 309)	Often	127	41.10
Physical activity	High	1063	33.72
(*n* = 3152)	Moderate	776	24.62
	Low	1313	41.66

Descriptive statistics from all participants following exclusions (*n* = 3152). Data are shown as frequency and percentage.

**Table 2 ijerph-20-06734-t002:** Baseline characteristics of subjects based on respective obesity categories.

Characteristic		Obesity Categories										*p*-Value
		Normal (*n* = 1005)		Overweight (*n* = 358)		Obesity (*n* = 603)		Central Obesity (*n* = 397)		Combination (*n* = 789)		
		*n*	%	*n*	%	*n*	%	*n*	%	n	%	
Sex (*n* = 3152)	Male	309	30.75	130	36.31	152	25.21	32	8.06	113	14.32	0.000 *
	Female	696	69.25	228	63.69	451	74.79	365	91.94	676	85.68	
Age (*n* = 3152)	19–44	606	60.30	259	72.35	552	91.54	36	9.07	138	17.49	0.000 *
	45–59	224	22.29	51	14.25	40	6.63	183	46.10	440	55.77	
	>60	175	17.41	48	13,41	11	1.82	178	44.84	211	26.74	
Education (*n* = 442)	Elementary	102	43.59	15	24.19	35	26.72	0	0	2	16.67	NS ^†^
	Junior high	13	5.56	5	8,06	10	7.63	0	0	0	0	
	Senior high	64	27.35	22	35.48	27	20.61	0	0	0	0	
	D1, D2, D3/university	55	23.50	20	32.26	59	45.04	3	100	10	83.33	
Income (*n* = 161)	<IDR 2 million	44	93.62	7	84.50	23	92	22	91.67	48	84.21	0.62 ^†^
	IDR 2–8 million	3	6.38	1	12.50	1	4	2	8.33	8	14.04	
	IDR 8-10 million	0	0	0	0	0	0	0	0	0	0	
	>IDR 10 million	0	0	0	0	1	4	0	0	1	1.75	
Omega-3 fatty acid intake (*n* = 3152)	Adequate	30	2.99	8	2.23	21	3.48	25	6.30	33	4.18	0.02 *
	Low	975	97.01	350	97.77	582	96.52	372	93.70	756	95.82	
Fast food intake (*n* = 309)	Seldom	43	59.72	31	59.62	62	65.26	13	50	33	51.56	0.42 *
	Often	29	40.28	21	40.38	33	34.74	13	50	31	48.44	
Physical activity (*n* = 3152)	High	419	41.69	151	42.18	253	41.96	172	43.32	318	40.30	0.96 *
	Moderate	243	24.18	95	26.54	149	24.71	94	23.68	195	24.71	
	Low	343	34.13	112	31.28	201	33.33	131	33.00	276	34.98	
**hs-CRP (mg/dL)**												
		0.05 (0.00–4.62)		0.07 (0.00–4.90)		0.16 (0.00–3.21)		0.09 (0.00–2.86)		0.17 (0.00–3.79)		

Descriptive statistics from all participants following exclusions (*n* = 3152) and by obesity categories. * = *p*-value of Pearson’s chi-squared analysis. † = *p*-value of Fisher’s exact test analysis. NS = not significant. Data are shown as frequency and percentage except for hs-CRP values, which are shown as median (min.–max.).

**Table 3 ijerph-20-06734-t003:** Baseline characteristics of subjects based on omega-3 fatty acid intake.

Characteristics		Omega-3 Fatty Acid Intake				*p*-Value
		Adequate		Low		
		n	%	n	%	
Sex	Male	32	27.35	704	23.20	0.30 *
(*n* = 3152)	Female	85	72.65	2331	76.80	
Age (*n* = 3152)	19–44	54	46.15	1537	50.64	0.18 *
	45–59	32	27.35	906	29.85	
	≥60	31	26.50	592	19.51	
Education (*n* = 442)	Elementary	3	15.79	151	35.70	1.00 ^†^
	Junior high	0	0	28	6.62	
	Senior high	5	26.32	108	25.53	
	D1, D2, D3/university	11	57.89	136	32.15	
Income (*n* = 161)	<IDR 2 million	7	100	137	88.96	0.13 ^†^
	IDR 2–8 million	0	0	15	9.74	
	IDR 8–10 million	0	0	0	0	
	>IDR 10 million	0	0	2	1.30	
Obesity categories	Normal	30	25.64	975	32.13	0.02 *
(*n*= 3152)	Overweight	8	6.84	350	11.53	
	Obesity	21	17.95	582	19.18	
	Central obesity	25	21.37	372	12.26	
	Combination	33	28.21	756	24.91	
Fast food intake (*n* = 309)	Seldom	8	50	174	59.39	0.46 *
	Often	8	50	119	40.61	
Physical activity (*n* = 3152)	High	51	43.59	1262	41.58	0.67 *
	Moderate	31	26.50	745	24.55	
	Low	35	29.91	1028	33.87	
hs-CRP (mg/dL)						
		0.12 (0.00–3.38)		0.10 (0.00–4.90)		

Descriptive statistics from all participants following exclusions (*n* = 3152) and by omega-3 fatty acid intake. * = *p*-value of Pearson’s chi-squared analysis. † = *p*-value of Fisher’s exact test analysis. Data are shown as frequency and percentage except for hs-CRP value’s which are shown as median (min.–max.).

**Table 4 ijerph-20-06734-t004:** Comparison of hs-CRP based on obesity categories.

Obesity Categories	hs-CRP (mg/dL)	*p*-Value
Normal	0.05 (0.02–0.14)	0.0001
Overweight	0.08 (0.03–0.18)	
Obesity	0.16 (0.07–0.36)	
Central obesity	0.09 (0.04–0.21)	
Combination	0.17 (0.07–0.37)	

**Table 5 ijerph-20-06734-t005:** Pairwise comparison of hs-CRP based on obesity categories.

Comparison between Obesity Categories	*p*-Value
Normal and overweight	0.0001
Normal and obesity	0.0000
Normal and central obesity	0.0000
Normal and combination	0.0000
Overweight and obesity	0.0000
Overweight and central obesity	0.04
Overweight and combination	0.0000
Obesity and central obesity	0.0000
Obesity and combination	0.43
Central obesity and combination	0.0000

**Table 6 ijerph-20-06734-t006:** Comparison of hs-CRP based on omega-3 fatty acid intake levels.

Omega-3 Fatty Acid Intake	hs-CRP (mg/dL)	*p*-Value
Adequate	0.12 (0.04–0.25)	0.93
Low	0.10 (0.04–0.26)	

Table 6 shows statistical analysis by Mann–Whitney U test. Data are shown as median (min.–max.). The hs-CRP values based on omega-3 fatty acid intake levels yielded no significant difference.

## Data Availability

Restrictions apply to the availability of these data. All IFLS 5 public release data and documentation include the overview/field report, user guide, and data file codebooks. Data from RAND IFLS 5 are available at https://www.rand.org/well-being/social-and-behavioral-policy/data/FLS/IFLS/access.html, accessed on 15 January 2023, after agreeing to several conditions and registering an email address.

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
