# Peer review of "Comparison of hs-CRP in Adult Obesity and Central Obesity in Indonesia Based on Omega-3 Fatty Acids Intake: Indonesian Family Life Survey 5 (IFLS 5) Study"

_ijerph, 2023, doi:10.3390/ijerph20186734_

Round 1

Reviewer 1 Report

This study compared high-sensitivity CRP (hs-CRP) in adult obesity and central obesity in Indonesia based on omega-3 fatty acids intake using the Indonesian Family Life Survey (IFLS). It is known that inflammation parameters are elevated in obese people, and it is also known that omega-3 fatty acids regulate lipid metabolism. A good thematic approach, with multifactorial analyses, to the influence of the external environment and the CRP relationship in relation to one population group is interesting if it is known that the Mediterranean diet is the best diet

The article is well written, easy to read, and follow the data, which are excellently tabulated

The conclusion is absolutely in accordance with the topic of the work and research and deals with the central question

References adequately accompany the article

Author Response

Dear Reviewer 1,

We would like to thank you for your feedback, attention, review, and appreciation of our work on the manuscript entitled "Comparison of hs-CRP in adult obesity and central obesity in Indonesia based on Omega-3 Fatty Acids intake: Indonesian Family Life Survey 5 (IFLS 5) study" (Manuscript ID: ijerph-2563405).

We hope that this manuscript can continue to be processed.

Kind regards

Ginna Megawati

Faculty of Medicine

Universitas Padjadjaran Bandung-Indonesia

Reviewer 2 Report

Authors have compared the CRP level in obesity with reference to Omega-3 Fatty Acids intake from the data of Indonesian socioeconomic and health longitudinal survey done in 2014–2015. Well written and clear presentation of old survey data. Below is my comment.

1. Data was obtained from RAND IFLS 5 and are available at https://www.rand.org/well-being/social-and-behavioral policy/data/FLS/IFLS/access.html after agreeing to several conditions and registering an email address... Double check of ethical concern and conflict of interest.. Anyone can have access of this survey data just by accepting the term and conditions and can reinterpret the same data for publication purpose?

2. Improve the discussion section with reference to relevant recent work done. Why only CRP and Omega-3 Fatty Acids intake were important parameters for consideration in obesity?

3. Share the sample of IFLS 5 household questionnaires for review purpose.

4. Rewrite the conclusion and mention the outcome from co relation between the different parameters used in this Omega-3 Fatty Acids intake based obesity study.

5. Add a figure to represent the co-relation study between Age, Sex and Omega-3 Fatty Acids intake in obesity.

Author Response

Dear Reviewer 2,

We would like to thank you for your feedback on our manuscript entitled "Comparison of hs-CRP in adult obesity and central obesity in Indonesia based on Omega-3 Fatty Acids intake: Indonesian Family Life Survey 5 (IFLS 5) study" (Manuscript ID: ijerph-2563405).

We hope that the improvements we make meet your expectations. This manuscript improvement also considers the opinions and suggestions of other reviewers. We hope that this manuscript can continue to be processed.

Kind regards

Ginna Megawati

Faculty of Medicine

Universitas Padjadjaran Bandung-Indonesia

Reviewer 3 Report

Dear author;

The article needs corrections in terms of statistical analyzes and discussions. I have stated these in my attached report. If these corrections are made, it may be appropriate to publish your article in the journal.

Respects.

Author Response

Dear Reviewer 3,

We would like to thank you for your feedback on our manuscript entitled "Comparison of hs-CRP in adult obesity and central obesity in Indonesia based on Omega-3 Fatty Acids intake: Indonesian Family Life Survey 5 (IFLS 5) study" (Manuscript ID: ijerph-2563405).

We hope that the improvements we make meet your expectations. This manuscript improvement also considers the opinions and suggestions of other reviewers. We hope that this manuscript can continue to be processed.

Kind regards

Ginna Megawati

Faculty of Medicine

Universitas Padjadjaran Bandung-Indonesia

Round 2

Reviewer 3 Report

Consideration and implementation of the recommendations were appropriate for the publication of the study.

Thank you for your hard work.